# Obstructive Sleep Apnea in Neonates

**DOI:** 10.3390/children9030419

**Published:** 2022-03-15

**Authors:** Indira Chandrasekar, Mary Anne Tablizo, Manisha Witmans, Jose Maria Cruz, Marcus Cummins, Wendy Estrellado-Cruz

**Affiliations:** 1Division of Neonatology, Department of Pediatrics, Valley Children’s Hospital, Madera, CA 94305, USA; 2Division of Pulmonary and Sleep Medicine, Valley Children’s Hospital, Madera, CA 94305, USA; mtablizomd@gmail.com or; 3Department of Pediatrics, Stanford University School of Medicine, Stanford, CA 94305, USA; 4Stollery Children’s Hospital, Edmonton, AB T6G 2B7, Canada; manishawitmans@gmail.com; 5Department of Pediatrics, Children’s Mercy Hospital, Kansas City, MO 64108, USA; jccruz@cmh.edu; 6School of Medicine, University of California San Francisco, Fresno, CA 94143, USA; marcus.cummins@ucsf.edu

**Keywords:** neonates, infants, obstructive sleep apnea, upper airway obstruction

## Abstract

Neonates have distinctive anatomic and physiologic features that predispose them to obstructive sleep apnea (OSA). The overall prevalence of neonatal OSA is unknown, although an increase in prevalence has been reported in neonates with craniofacial malformations, neurological disorders, and airway malformations. If remained unrecognized and untreated, neonatal OSA can lead to impaired growth and development, cardiovascular morbidity, and can even be life threatening. Polysomnography and direct visualization of the airway are essential diagnostic modalities in neonatal OSA. Treatment of neonatal OSA is based on the severity of OSA and associated co-morbidities. This may include medical and surgical interventions individualized for the affected neonate. Based on this, it is expected that infants with OSA have more significant healthcare utilization.

## 1. Introduction

OSA is caused by partial (hypopnea) or complete (apnea) obstruction of the upper airway during sleep, and neonates are also at risk for this. These respiratory events are often associated with intermittent hypercapnia and hypoxemia, cortical arousals, and sleep architecture disturbances.

Neonates have distinctive anatomic and physiologic characteristics that predispose them toward recurrent airway obstruction and gas exchange abnormalities. Neonates have a superiorly positioned larynx, a highly compliant ribcage, reduced airway stiffness, immature ventilatory control, and craniofacial features that narrow the airway lumen. All of these features can predispose the neonates to upper airway obstruction and gas exchange abnormalities [1]. Additionally, anatomical narrowing from the nose to the larynx caused by congenital or acquired abnormalities can contribute to increased upper airway resistance which further predisposes neonates to airway obstruction, especially during sleep. Neonates may also have multiple levels of obstruction and multiple etiologies, especially in those with craniofacial malformations, making the diagnosis and management more complex.

## 2. Prevalence

Pediatric OSA has an estimated prevalence of 1−5%, but the prevalence in neonates remains unknown. The lack of consensus over diagnostic criteria and limited normative data for OSA in infants contribute to the difficulty in establishing its true prevalence. Neonatal OSA has been reported to occur more frequently in premature babies [2], in the presence of craniofacial malformations, neuro-muscular disorders, exposure to prenatal smoking [3], bronchopulmonary dysplasia [4], males [5], and obese neonates [6].

## 3. Pathophysiology

There are many factors that predispose neonates to have obstructive sleep apnea. One of the essential features of neonates is increased upper airway resistance during sleep compared to older children and adults. Additionally, neonates have a compliant chest wall which contributes to decreased functional residual capacity, reduced airway stiffness, and immature central control of breathing [1]. They also spend more time sleeping with proportionately increased time in REM (rapid eye movement), during which respiratory muscles are atonic except for the diaphragm [7]. Other factors such as neck flexion, reduced ability to clear airway secretions, and increased frequency of gastroesophageal reflux can also worsen predisposition of OSA in neonates [8]. Table 1 shows the difference in anatomy of neonates vs. older children along with its implication in causing airway obstruction.

Upper airway narrowing from the nasal area down to the trachea can lead to OSA in neonates. The common anatomical features of the upper airway leading to OSA include nasal or choanal stenosis, midface hypoplasia, underdeveloped jaw (micrognathia), large tongue, laryngeal web, floppy epiglottis (laryngomalacia), and narrowed trachea (subglottic stenosis or tracheomalacia). Infants with neurologic abnormalities who developed OSA in the neonatal period are those with abnormal upper airway tone, as seen in neuromuscular disorders (spinal muscular atrophy), immature and abnormal control of breathing, and abnormal brain anatomy (Arnold Chiari malformation). Table 2 shows conditions that are commonly associated with OSA in neonates.

### 3.1. Prematurity

Prior studies have shown an association between premature birth (gestational age (GA) < 37 weeks at birth) and increased risk for OSA [9,10]. The narrower, shorter and collapsible airway, abnormal craniofacial characteristics (facial asymmetry and dolichocephaly), and impaired neuromuscular coordination of the upper airway are possible reasons for premature babies to be at risk for OSA. The need for prolonged respiratory support for extremely premature (GA < 28 weeks) babies can also increase the risk of OSA. Prolonged intubation can alter the shape and growth of oropharyngeal cavity, resulting in a narrow and high arched palate along with the risk of developing subglottic stenosis. In addition, prolonged noninvasive respiratory support using nasal interface can cause midface hypoplasia and contribute to upper airway obstruction.

Jaleel et al. evaluated 89 premature babies for sleep-related breathing disorder. Twenty-three (25.8%) were born preterm (GA < 32 weeks) and nine (10.1%) were extremely preterm (GA < 28 weeks). Findings on polysomnography suggested an association between premature birth and markedly increased AHI (mean (SD): 6.5 (9.8) compared to term babies (mean (SD): 4.6 (6.4), *p*  <  0.05). This finding persisted when adjusted for maternal age and birth weight (*p*  < 0.05). When the group was subdivided based on GA, they found a higher rate of OSA in babies with lower GA [11].

There is decreased respiratory effort throughout an obstructive event reported in premature infants making obstructive events worse in premature babies [12]. A robust neuromuscular reflex activation of the upper airway during airway obstruction has been seen among term infants during airway obstruction, a response which may be absent in preterm infants.

### 3.2. Craniofacial Abnormalities

Abnormal facial development in infants can cause upper airway obstruction. The facial abnormalities that are commonly associated with OSA are maxillary and mandibular hypoplasia, micrognathia, and retrognathia. Most of these facial skeletal abnormalities are generally part of a syndrome and are associated with underlying genetic disorders. Children with severe craniofacial abnormalities may present with symptoms of upper airway obstruction at birth.

Robin sequence (RS) is characterized by a triad of micrognathia, with resulting glossoptosis and subsequent upper airway obstruction. There is often an associated cleft palate. The obstruction arises from crowded oropharyngeal space causing obstruction at the tongue base. Patients with RS tend to have OSA, but in severe cases, it can present with significant airway obstruction in the newborn period that can be life-threatening. In some cases, OSA can worsen when the cleft palate is corrected. The risk of OSA persists in these affected infants throughout their life. 

In other craniofacial syndromes, the airway obstruction can be variable and may occur at multiple levels. Apert syndrome, for example, is characterized by choanal atresia, congenital bony nasal stenosis, deviated septum, thick long soft palate, and abnormalities involving the trachea and bronchial airways. A combination of these craniofacial abnormalities often makes it impossible to manage the airway without early surgical intervention, including tracheostomy.

### 3.3. Laryngeal Abnormalities

Laryngomalacia is one of the most common etiologies of inspiratory stridor and OSA in infants. Upper airway obstruction is often related to the collapse of the epiglottis and arytenoid cartilages during inspiration. Airway obstruction often gets worse when they are in supine position, active, eating, or sometimes asleep. Infants with severe laryngomalacia can present with severe OSA, hypoxemia, chest retractions, inability to suck and swallow because of the challenges related to eating and breathing leading to failure to thrive, and may require surgical intervention. In addition, some may have concomitant tracheo-bronchomalacia. Gastroesophageal reflux is also commonly noted with laryngomalacia and can further worsen airway obstruction.

### 3.4. Neurological/Neuromuscular

Neonates with neuromuscular disorder (NMD), Trisomy 21, Prader–Willi syndrome, and mitochondrial disorders can present with generalized poor muscle tone, leading to airway collapsibility and subsequent OSA. Hypotonia seen during the neonatal period in certain conditions may subsequently improve with age. Neonates with severe cerebral palsy also present with abnormal upper airway muscle tone, which often is associated with laryngeal dystonia, severe laryngomalacia, and concurrent pseudobulbar palsy, all of which can lead to OSA. Affected neonates with OSA may suffer from long-term consequences, hence, early recognition and appropriate management should be initiated.

In neonates with neuromuscular disorder, the poor tone can be progressive in some cases that can result in a lifelong need for respiratory support. In general, a reduction in functional residual capacity is seen in the supine position and during sleep. This lung volume reduction can be accentuated during periods of muscle atonia [13,14,15]. In patients with neuromuscular disorders, OSA results from a combination of chest wall stiffness and poor upper airway tone from respiratory muscle weakness [15]. Chronic and progressive hypoventilation, as seen in many of these disorders, can lead to blunted hypercapnic and hypoxic responses. In turn, this can lead to compensatory mechanisms (e.g., retention of bicarbonate by the kidney) that can further decrease responsiveness to gas exchange abnormalities and progressive worsening of ventilation [16,17].

### 3.5. Other Upper Airway Abnormalities Associated with OSA

Choanal atresia is a congenital abnormality characterized by narrowing of the posterior nasal airway causing OSA. Affected newborns can present with significant respiratory distress requiring immediate medical and/or surgical intervention. This condition is often associated with genetic conditions including Crouzon, Treacher–Collins, and CHARGE syndromes. The obstruction may be classified as bony (30%) or mixed membranous/bony (70%) [18]. Choanal atresia may be bilateral (55%) or unilateral (45%) [19]. Bilateral choanal atresia often presents at birth with symptoms of airway obstruction. On the other hand, unilateral choanal atresia may not present until later in infancy with nasal congestion, nasal obstruction, and disturbed sleep. Flexible fiberoptic endoscopy may be helpful but definitive diagnosis is established by computerized tomography. Surgical treatment of choanal atresia is highly effective.

Congenital and acquired subglottic stenosis (SGS) is another etiology for neonatal OSA. The narrowest portion of the upper airway in infants is the subglottic area at the level of the cricoid cartilage. Congenital SGS is a result of developmental defects of airway cartilage. This is usually associated with other genetic syndromes. Acquired SGS can result from prolonged intubation, trauma to the airway, infection, and chronic inflammation such as from severe GERD. This condition is associated with swelling of the soft tissues, hypertrophy or narrowing of the cartilage with a subglottic diameter of less than 4 mm in a term infant. Affected infants often present with varying degrees of respiratory distress and biphasic stridor. Most infants with mild disease often will not require surgical intervention as symptoms gradually improve as the larynx grows. Medical management is often indicated in the presence of significant gastroesophageal reflux (GERD). Infants with significant airway obstruction may need supportive treatment. These patients may present with severe OSA and considerable respiratory distress during sleep. Treatment is individualized with the expectation of an airway that adequately supports the growing infant.

Cleft palate may occur as an isolated anomaly (30%) or arise as part of a genetic syndrome (70%) [20]. The defect is associated with a smaller posterior airway space [21] and an increased risk of OSA [22]. Nonsyndromic cleft palate may also be associated with some degree of maxillary hypoplasia. Surgical repair of the cleft palate may result in impaired anteroposterior growth of the maxilla that may predispose affected children towards OSA in the future [23]. Approximately 20% of children with a cleft palate developed velopharyngeal insufficiency and subsequently will require pharyngoplasty or pharyngeal flap, which further increases the risk of OSA [24,25]. Cognitive impairments have been observed in children with cleft palate, which is presumed to be from eustachian tube dysfunction and longstanding untreated OSA.

## 4. Diagnosis

Multidisciplinary evaluation, which includes neonatology, pulmonology, otolaryngology, plastic surgery, and genetics, is crucial when evaluating neonates with suspected OSA to establish the diagnosis and determine the optimal plan of treatment. Other specialists such as gastroenterologists and neurologists may also need to be involved. Often, speech and language pathologists are required to assist with feeding modifications because of the compromised airway and risk of aspiration with feeding.

### 4.1. Clinical Presentation

The clinical presentation of neonatal OSA in the immediate post-natal period is highly variable. Neonates with severe craniofacial abnormalities may have clinically apparent signs of respiratory distress as increased work of breathing, including observable apneic episodes, tachypnea, suprasternal retractions, even head bobbing, stridor or overt respiratory failure. Neonates who are less severely affected may simply present with noisy breathing, snoring or may have no respiratory symptoms at all. Feeding difficulties, choking while eating, and poor weight gain due to increased metabolic demands from upper airway obstruction may also be observed. Fragmented or disrupted sleep with gasping and frequent arousals may also be noted. Some infants may present with prolonged sleepy episodes where they have to be woken up to feed.

### 4.2. Physical Examination

Physical examination findings of neonates with OSA can be highly variable depending on the etiology, and this is further complicated as some symptoms may be state-dependent and only be present when the infant is asleep or only during wakefulness. The severity of upper airway obstruction may also vary greatly in severity. Patients with craniofacial abnormalities can have maxillary hypoplasia, micrognathia, and retrognathia. Neonates with neuromuscular disorders that present in the neonatal period such as Spinal. Muscular Atrophy (Type 1) may have clinically apparent hypotonia that can lead to respiratory and feeding difficulties. Increased work of breathing is noted at rest, and worsening respiratory distress can occur when the infant is stressed or feeding. Swallowing dysfunction may also be present as a result of the tachypnea or occur secondarily from significant airway obstruction. Hemangiomas that are noted on the face or in the beard distribution can be associated with airway hemangiomas, thus contributing to airway obstruction.

### 4.3. Airway Endoscopy

Fiberoptic evaluation of the upper airway is essential prior to surgery to identify the site of airway obstruction and to evaluate for other causes of obstruction. It is not uncommon for infants to have multiple sites of airway obstruction. For example, choanal atresia may be present with laryngomalacia and/or tracheomalacia. Airway endoscopy allows direct visualization of the airway, localize the level of airway obstruction, and assessment of airway dynamics. Micro-laryngoscopy-bronchoscopy is useful to confirm the presence of glossoptosis, evaluate subglottic structures, and to rule out other causes of upper and lower airway obstruction.

### 4.4. Imaging Studies

Imaging studies like lateral neck radiographs may be useful in assessing the degree of micrognathia. Computed tomography (CT) can be used to assess the lengths and volumes of airway and bony structures. Members of the International Pediatric Otolaryngology Group (IPOG) have agreed that high-resolution computerized tomography (HRCT) of nasal cavity is the gold standard in the diagnosis of choanal atresia in the 2019 consensus [26]. Although imaging studies may provide an understanding of the degree of upper airway compromise, it should not replace direct visualization of the airway.

### 4.5. Polysomnography (PSG)

Attended polysomnography is the gold standard for the diagnosis of neonatal OSA. PSG can document the different forms of apnea, OSA severity, associated gas exchange abnormalities, and provide valuable details for treatment planning and subsequent response to treatment. However, this test is also not a substitute for direct visualization of the airway. A typical montage includes electroencephalographic (bilateral central, frontal and occipital) channels, electromyography (left and right anterior tibial and chin), bilateral electrooculography (EOG), pulse oximeter and pulse waveform, nasal pressure transducer, oronasal airflow thermistor, end-tidal capnography or transcutaneous carbon dioxide, chest and abdominal inductance plethysmography, body position sensor, microphone, and real-time synchronized video monitoring.

The current AASM guidelines recommend different rules for sleep staging for those less than 2 months post term referred to as infants and those older than 2 months post term to 18 years of age referred to as children) [27].

The criteria for scoring respiratory events in PSG in neonates are the same as the one used in pediatric patients up to 18 years. Apneas are scored similar to older children where there is at least a ≥90% drop in peak signal excursion from the pre-event baseline using an oronasal thermal sensor. The apnea event is considered obstructive when the event lasts for at least 2 respiratory cycles during the baseline breathing and is associated with the presence of persistent or increased respiratory effort during the time that there is absent airflow. Obstructive hypopneas in infants are similarly scored as older children, with criteria being at least a ≥30% drop in nasal pressure signal compared to the pre-event baseline, and there must be at least a 3% drop in oxygenation desaturation from the pre-event baseline or the event is associated with an arousal. The duration of the 30% drop in nasal cannula flow for pediatric patients must be for at least 2 breaths [27].

The AASM recommends monitoring C02 either by TcC02 or EtC02 during baseline study and is optional during PAP titration study. A patient is said to have obstructive hypoventilation during sleep if the C02 is ≥50 mm Hg for at least 25% of total sleep time and associated with snoring or if there are signs of increased upper airway resistance manifested as paradoxical breathing and flattening of inspiratory nasal pressure waveform [27].

### 4.6. Severity of OSA Based on PSG

Diagnostic criteria for neonatal OSA are based on pediatric criteria. Normative data for infants and neonates have not been well established.

The severity scale of OSA used in neonates in some centers is based on older children (AHI of ≥1 to <5 is considered mild OSA, AHI of ≥5 to <10 moderate OSA, and an AHI > 10 is severe OSA).

## 5. Treatment

Treatment of neonatal OSA requires a multidisciplinary collaboration because of the complexity of care that the infant needs. Definitive management will depend on the severity of respiratory and feeding difficulties, neurocognitive dysfunction, and associated genetic syndrome. Neonates who present with considerable upper airway obstruction at birth, profound respiratory distress, significant gas exchange abnormalities, failure to thrive, or significant craniofacial malformation require more aggressive therapy. Early surgical intervention should be considered in neonates with airway compromise. The respiratory problems in these cases are not isolated during sleep but may also occur while awake.

In the majority of neonates with suspected OSA, the decision making may not be as clear because of the rapid growth of the infant. In addition, the clinical presentation can change dramatically week to week. The treatment decision is generally guided by the following five factors and their associated degree of impairment: sleep, breathing, feeding, growth, and development. The treatment options range widely, from conservative management such as continued observation and/or prone positioning for mild cases of OSA, to more targeted types of surgery including supraglottoplasty, mandibular distraction osteogenesis (MDO), and tongue–lip adhesion (TLA). In the presence of severe OSA from multilevel obstruction, a tracheostomy may be performed to entirely bypass the upper airway obstruction. As our understanding improves about the pathophysiology and causes of OSA, there is an emergence of different management options to avoid tracheostomy.

### 5.1. Non-Surgical Treatment

#### 5.1.1. Positive Airway Pressure (PAP) Therapy

PAP therapy is the recommended treatment in children with OSA who are not surgical candidates or where surgery alone may not be curative. The pressurized air delivered through a nasal or full-face interface creates a pneumatic stent to maintain upper airway patency and prevents upper airway collapse. Data on the use of PAP to treat neonatal OSA are limited and studies that are available are in older children. A single-center retrospective study compared PAP therapy in 41 infants (younger than six months) to 109 school-aged children (5–10 years). They found a median reduction in obstructive apnea-hypopnea index (OAHI) of 92.1% in infants, which was comparable to the 93.4% reduction observed in school-aged children. Furthermore, infants used PAP on 94.7% of nights, as compared to 83% among school-aged children [28]. These results suggest that PAP therapy in infants is well tolerated and effective. The use of PAP therapy in neonates may be quite challenging due to the limited available interfaces to fit neonates with craniofacial syndromes and concerns for the development of midface hypoplasia from prolonged PAP use. Given the limited data on PAP therapy in neonates, further research is needed to determine the efficacy and safety of PAP therapy in neonatal OSA.

#### 5.1.2. High-Flow Nasal Cannula (HFNC)

A recently proposed alternative therapy for OSA is the use of HFNC. There are limited studies for neonatal OSA with HFNC. A retrospective study of 10 infants (median age 34 weeks; IQR 27–38 weeks) using HFNC reduced the median obstructive apnea hypopnea index (OAHI) from 9.1 (IQR 5.1–19.3) to 0.9 (IQR 0–1.6; *p* < 0.005) events/h; median obstructive apnea index (OAI) from 5.8 (IQR 1.1–13.4) to 0 (IQR 0–0.9; *p* < 0.021) events/h; median obstructive hypopnea index (OHI) from 4.1 (IQR 0.9–6.8) to 0.1 (0–0.9; *p* < 0.017) events/h; and median oxygen saturation (SpO2) nadir increased from 88% (IQR 83–94%) to 94% (IQR 93–96%; *p* < 0.040) [29]. The efficacy of HFNC has been shown in several studies involving older children with OSA [30,31].

An interesting study using 3D airway models demonstrated that HFNC achieved sufficient distending pressure in 3D-printed airway models of pre-term and term neonates. This finding may explain the improvement reported on the use of HFNC in the treatment of OSA [32]. HFNC may be considered as an alternative treatment for infant and neonatal OSA. However, further studies are needed to establish the efficacy of HFNC in neonatal OSA in the home setting and its effects on the carbon dioxide levels in the presence of alveolar hypoventilation.

#### 5.1.3. Orthodontic Interventions

Various orthodontic devices can be used to alter the upper airway anatomy in neonates with OSA. These devices are less invasive than surgical options.

One of these devices is the pre-epiglottic baton plate (PEBP). This device opens the pharyngeal airway using a velar extension that moves the tongue in a more anterior position. The effect of PEBP in 122 children (4–42 days old) with isolated RS was evaluated using a sleep study from 2003 to 2012. They found a statistically significant reduction in the mixed-obstructive apnea index (MOAI) from 8.8 to 1.8 to 0.2, respectively. None of the infants required craniofacial surgery or tracheostomy following their PEBP treatment [33].

A prospective multicenter cohort study evaluated the efficacy of PEBP on 49 infants (< 1 year of age at admission) with RS. Within a mean hospitalization of 3 weeks, they showed a decrease in the MOAI from 15.9 (6.3–31.5) on admission to 2.3 (1.2–5.4) at discharge. At 3-month follow-up, the MOAHI decreased further to 0.7 (0.1–2.4) in the 32 infants. The desaturation index normalized (from 0.38 (0–2.7) to 0.0 (0–0.1) [34].

A 13-year, single-center study evaluated the use of PEBP in 132 newborns with clinical signs of RS (isolated RS, 111; syndromic RS, 21) from 2010 to 2019. Tracheostomy was performed in 5 of 21 patients with syndromic RS due to unsuccessful treatment with PEBP. All infants with isolated RS had improvement in breathing problems (desaturations and pre-sternal retractions) and all were discharged within an average of 8 days of PEBP therapy. None of these infants needed tongue–lip adhesion procedure or tracheostomy. There was no discontinuation or noncompliance rate. Polysomnography, however, was not performed as part of their study [35].

A narrative review on the use of the Tübingen palatal plate (TPP) approach to RS was presented by Poet and colleagues. Between 1998 and 2018, 443 infants with RS were treated in their center using TPP; in 129 of these infants, RS was part of an underlying syndrome. None of the 314 infants with isolated RS required a tracheostomy after initiation of treatment with the TPP. Of the syndromic patients, 23 (17%) ultimately required a tracheostomy, mostly those with laryngeal problems or swallowing disorder. The following findings were reported: (a) TPP was superior to a sham procedure in alleviating UAO in a randomized trial; (b) children who were treated with the TPP in infancy had an intellectual development within the reference range; (c) prone positioning was ineffective and associated with an increased risk of sudden death; (d) TPP decreased the MOAI to near-normal values, both in isolated RS and about 83% of syndromic RS, (e) 23 of 129 syndromic RS (5%) ultimately had a tracheostomy (f) data indicated that the TPP may induce mandibular catch-up growth, (g) TPP may help reduce respiratory complications following cleft closure in RS [36].

Treatment of upper airway obstruction in RS using palatal plates has been reported by various centers around the world. Current literature suggests that the use of palatal plate is a safe and effective method to treat neonatal OSA. Even though this approach is well studied, it is currently not widely applied internationally. Rapid-maxillary expansion (RME) is another orthodontic device that can be used to treat OSA. Orthodontic RME anchors a fixed device to opposing teeth using an expansion screw. The screw then gradually expands the device, thereby increasing the diameter of the hard palate over the course of treatment for several months [37]. To our knowledge, there are no studies on the use of RME in neonatal OSA.

#### 5.1.4. Supplemental Oxygen

The effects of supplemental oxygen in children with OSA have been previously reported. A recent meta-analysis conducted by Mehta and colleagues showed that oxygen therapy led to significant improvement in AHI and overall oxygenation. However, they also found that patients who received oxygen therapy had a longer average duration of apnea and hypopnea compared to placebo [38]. Data on the use of oxygen in neonates and infants with OSA are limited. In a study by Brockbank and colleagues using oxygen in 59 infants (13.0 ± 11.7 weeks and at O2-PSG was 15.4 ± 13.0 weeks) with OSA, they found a significant reduction in OAHI (19.7 ± 13 vs. 10.6 ± 11.7) and improved oxygenation without adverse effects on alveolar ventilation [15]. The use of supplemental oxygen may be considered as a treatment alternative for neonates who are not candidates for surgical intervention. Further studies, however, are needed to determine the efficacy and safety of this intervention in neonatal OSA.

#### 5.1.5. Pharmacologic Therapy

GERD and OSA often co-exist in the general population. To date, no studies have demonstrated a causal relationship between OSA and GERD. Although some studies have demonstrated a reduction in OSA severity using anti-GERD treatment in children [13,14], it does not definitely prove a causal relationship. A retrospective review of 126 neonates and infants (age 0–12 months) with OSA in a tertiary children’s hospital found that anti-GERD therapy was used commonly (69.8%) across all OSA severities (mild, moderate and severe) and it resulted in mean reduction of AHI of 45.5% [13]. The impact of anti-GERD treatment on childhood OSA has been studied, but there are no studies that are specifically in the neonatal population.

#### 5.1.6. Positional Therapy

Positional therapy, especially in children with micrognathia and glossoptosis, can affect the magnitude of upper airway obstruction during sleep. The supine position allows gravity to force the mandible and the tongue posteriorly, thereby increasing the degree of obstruction. It has been theorized that both prone and side-lying position may lead to improvement in OSA parameters. A retrospective review of 18 infants with PRS (mean age 44 ± 26 days) and severe OSA on PSG had compared sleep efficiency and OAHI in prone vs. supine positioning. They reported a statistically significant improvement in sleep efficiency and reduction in OAHI when infants were placed in a prone position. Interestingly, when each infant’s sleep parameters were analyzed individually, the prone position was best for 13 infants, and the supine position was better for 4 infants [39]. This study suggests that neonates should be only placed in a prone position on a case-by-case basis, depending on the response with position changes using respiratory and sleep outcome measures. A prospective study recruited 14 infants with RS (age 7–218 days) to undergo PSG in non-prone and prone sleep positions. They showed that the median OAHI was reduced from 16.0 to 14.0 from non-prone to prone sleep. The reduction in OAHI, however, was not statistically significant. The authors concluded that some infants with RS may benefit from prone positioning, but even in those infants with significant OSA improvement, airway obstruction was not fully resolved [40].

A larger study involving 76 infants with RS under 3 months of age reported that OSA was least severe in the side or prone positions, but most severe in the supine position. The median OAHI in the supine, side, and prone groups were 31, 16, and 19 per hour, respectively (*p* = 0.003). Another retrospective study of 27 infants with cleft palate ± cleft lip (age 1 month to 1 year) found no significant improvement in OSA metrics during non-supine sleep [41].

An ongoing large, multicenter randomized controlled SLUMBRS2 trial involving 244 infants (age 3–5 weeks) with cleft palate aims to evaluate the clinical effectiveness of back-lying versus side-lying sleep positioning in reducing oxygen desaturations resulting from OSA [42]. This trial will run for 36 months and will hopefully shed light on what would be a simple, safe, and cost-effective intervention to address OSA in infants with PRS and/or cleft palates. The body of evidence for sleep positioning in infants with OSA is clearly mixed and the aforementioned studies are not specific to the neonatal population.

#### 5.1.7. Environmental Modification

Tobacco, other indoor pollutants, and allergens can cause nasal congestion and upper airway inflammation, and can hence contribute to upper airway resistance and worsen OSA. Avoiding exposure to tobacco smoking and pollutants, especially in the neonates, will at least lessen chances of contributing to increased upper airway inflammation.

### 5.2. Surgical Management

#### 5.2.1. Mandibular Distraction Osteogenesis (MDO)

MDO is a surgical procedure that involves bilateral osteotomies of the mandibular rami, and placement of distraction apparatus bilaterally. The mandibular lengthening leads to advancement of the tongue base, thereby increasing the size of the upper airway. MDO is cost-effective and a more functional alternative than tracheostomy for treatment of RS.

MDO is considered the first-line surgical therapy for treating severe RS. It primarily targets the hypoplastic mandible, which is the primary abnormality in RS. Long-term mandibular growth disturbance has been reported in infants with syndromic micrognathia. Affected infants who are not expected to have normal postnatal mandibular growth will likely benefit from MDO. Airway obstruction may worsen in infants with syndromic micrognathia, especially in the first year of life. Long-term follow-up studies by Bull et al. indicate that 65% of these infants have either chronic snoring or OSA [43]. Most infants had resolution of their OSA within a few weeks after mandibular distraction. Long-term follow-up indicates that 80% of these infants remain free of OSA [44].

Several small studies have looked at the efficacy of MDO in the treatment of OSA due to severe micrognathia. A study of 10 infants and older children showed that MDO provided a consistent change in the position of the tongue base which led to improvement in upper airway obstruction by increasing the measured effective airway spac [45]. A single surgeon retrospective review of non-syndromic neonates with RS were treated with MDO (*n* = 24) or TLA (*n* = 15). The following outcomes were evaluated: length of intensive care unit stay, time of extubation, incidence of tracheostomy, and surgical complications. Changes in oxygen saturation and apnea hypopnea index (AHI) were included in the polysomnography data analysis which were obtained at 1 month and 1 year postoperatively. They reported no post procedure tracheostomies in the MDO group vs. 4 in the TLA group (76.5% vs. 82%; *p* < 0.05). Preoperatively, oxygen saturations were significantly lower in the MDO group compared with the TLA group (76.5% vs. 82%; *p* < 0.05). AHI was significantly higher in the MDO group compared with the TLA group (47 vs. 37.6; *p* < 0.05). Post operatively, they found that patients who underwent MDO had significantly lower AHI and higher oxygen saturations at 1 month and 1 year (*p* < 0.05). When they looked at non syndromic patients with RS, they found that MDO had superior outcome measures in terms of AHI, oxygen saturation, and incidence of tracheostomy as compared to TLA [46].

One prospective study of 17 infants (5–120 days) with severe micrognathia and OSA refractory to conservative therapy underwent MDO. Fourteen out of 17 patients were successfully extubated, while the remaining 3 patients required tracheotomy due to persistent respiratory compromise (2) and persistent upper airway obstruction (1). In 10 of 17 infants who had both pre- and post-operative PSGs, a 55% improvement in OAHI was noted [47].

A retrospective review was conducted on 28 patients (age 11–312 days) with laryngoscopy-confirmed tongue-base airway obstruction who underwent MDO. Pre-operative PSGs obtained from 20 patients revealed an average AHI of 39.3 ± 22.0/h. Post-operative PSGs obtained from 14 of these 20 patients revealed a statistically significant reduction in AHI to 3.0 ± 1.5/h. There were 4 patients reported as therapeutic failures, 3 patients had minor surgical complications, and 1 had major surgical complications [43]. A retrospective, 14-year, single-institution study of 31 neonates (5–34 days old) with RS and severe OSA at baseline (defined as pre-operative OAHI ≥ 10) underwent MDO. The authors compared the respiratory parameters and sleep architecture before versus after surgery. The mean age was 13 days (5 to 34 days) at preoperative polysomnography and 80 days (50 to 98 days) at postoperative polysomnography. They found a significant reduction in OAHI 38.3 (23.4 to 61.8) preoperatively versus 9.4 (5.3 to 17.1) postoperatively along with significant improvement in sleep efficiency and oxygen saturation nadir. Although 26 neonates had a 50 percent reduction in OAHI postoperatively, all neonates had residual OSA, with 15 of the 26 having persistent severe OSA following surgery. [48].

A recent meta-analysis conducted by Ow and Cheung showed that MDO was effective in preventing tracheostomies in 91.3% of infants, as well as relieving OSA symptoms in 97% of these infants [49].

In a retrospective cohort study that compared TLA and MDO in patients with RS and OSA, patients who underwent TLA had an average age of 28.2 ± 23.1 days at the time of surgery, compared to 87.1 ± 81.7 days for infants who underwent MDO. MDO, however, was found to be significantly more effective than TLA, with a 94.6% decrease in post-operative AHI and OSA severity score, compared to a 33.5% decrease in infants that underwent TLA [50]. Even though TLA can be performed earlier and is associated with less morbidity, MDO has shown to be far more effective [44,45,46].

Cheng and colleagues reported moderate-to-severe residual OSA after MDO and TLA in a series of RS infants with comorbid airway lesions including choanal atresia, epiglottal abnormalities, laryngomalacia, and tracheal stenosis [51].

Complications from MDO include infection, facial nerve injuries, temporo–mandibular joint dysfunction, hypertrophic scarring, and disruption of tooth buds [52,53]. Long-term effects of this procedure are not yet known.

#### 5.2.2. Supraglottoplasty

Surgical treatment of laryngomalacia in infants with OSA from severe laryngomalacia has evolved away from tracheostomy in favor of supraglottoplasty. The use of endoscopic techniques has resulted in reduced morbidity and resolution of OSA in the majority of infants [54,55]. Supraglottoplasty is a surgical procedure that includes removal of redundant soft tissue overriding the accessory cartilages, incision of the aryepiglottic folds to release the epiglottis, and trimming the lateral edges of the epiglottis. Airway evaluation using direct laryngoscopy and bronchoscopy must be performed prior to endoscopic operative procedure to fully evaluate the airway to rule out the presence of any other airway lesions. Complications reported after supraglottoplasty are rare. Infants usually require 1–2 days of hospitalization.

Supraglottoplasty has been shown to improve total sleep time, decrease obstructive events, and improve oxygenation [54,55]. Although the majority of infants who were followed-up at about 3–5 months after surgery improved after supraglottoplasty, significant residual OSA may persist [54,55], particularly in the presence of comorbid conditions including tonsillar hypertrophy, micrognathia, tracheo-bronchomalacia, micrognathia, and neuromuscular disease. [56,57]. Longitudinal outcomes following this surgery are not known.

A large, systematic review of several online medical databases examined the available literature on supraglottoplasty outcomes. Among nine studies that met the inclusion criteria, the mean age at the time of surgery was 6.5 months [58]. In a retrospective review of 20 children with laryngomalacia who underwent supraglottoplasty at a tertiary center, the youngest child was 2 weeks old, and the mean age was 3.9 months. They reported statistically significant reductions in postoperative AHI and obstructive apnea index. The only reported complication was dysphagia, occurring in five infants, but this resolved within one month for each of them [59]. The long-term implications of surgery on the impact of long-term OSA, however, has not been studied.

#### 5.2.3. Tongue-Lip Adhesion (TLA)

TLA is another surgical intervention that can relieve airway obstruction related to glossoptosis. This surgical procedure aims to prevent the tongue from occluding the airway by suturing the tip of the tongue to the lower lip to anteriorly protract the tongue [60]. The effectiveness of TLA on infants and neonatal OSA has not been extensively studied. Bijnen and colleagues reported clinical improvement in OSA in 60% of infants (1–98 days old at presentation) with isolated tongue-base obstruction following TLA [61]. Sedaghat and colleagues found a reduction in AHI after TLA in 8 infants (mean age 6 days pre operatively) with RS [62]. Preoperatively, 7 of 8 infants had severe OSA. Postoperatively, 1 infant had resolution of OSA, 2 had mild OSA, and 2 had moderate OSA.

A retrospective study evaluated the outcomes of 37 infants (8 to 210 days) with RS with a mean age of 45 days who underwent TLA from 2004 to 2015. All patients had severe OSA and required respiratory support (intubation, non-invasive positive pressure ventilation, or nasopharyngeal airway) prior to the surgery. They demonstrated statistically significant improvements in oxygen saturation, hypercapnia, AHI, and bradycardia on postoperative PSG. However, 8 patients (21.6%) had persistent severe OSA and required tracheostomy or noninvasive ventilation [60]. Another retrospective study evaluated 18 infants (mean age 28 days) with RS who underwent TLA from 2011 to 2014. While oxygen saturation nadir, OSA severity and arousals per hour all showed statistically significant improvements, only 9 patients (50%) met the criteria for a successful outcome (defined as post operative AHI of <5/h with no additional intervention) [63].

#### 5.2.4. Tongue Reduction Surgery

Tongue reduction surgery can significantly reduce the degree of upper airway obstruction among infants with Beckwith–Wiedemann syndrome who experience OSA due to severe macroglossia. A 5-year single-center retrospective study looked at 36 children (mean age of 9.5 months) with Beckwith–Wiedemann who underwent tongue reduction surgery. They found a statistically significant reduction in OAHI from 30.9 ± 21.8/h to 10.0 ± 18.3/h [64]. To our knowledge, there is no existing data on the use of tongue reduction surgery in neonates.

#### 5.2.5. Choanal Atresia Repair

Standard repair options for choanal include puncture, dilatation, ablation, and drilling of the atretic plate. Post-operative stenting, which is a part of some of these surgical techniques, may promote granulation and scarring, with possible progression to restenosis. To date, five surgical techniques for choanal atresia repair have been described: transpalatal, transnasal, sublabial transnasal, transantral, and transseptal approaches [65]. With the advent of endoscopic instruments and techniques, the transnasal approach has been favored by most surgeons and is the preferred technique recommended by the IPOG [26].

The use of stenting and mitomycin C as an adjunct therapy to prevent restenosis are a controversial topic in the management of choanal atresia as there is no clear evidence to support the effectiveness of using stents or mitomycin after choanal atresia repair. Analysis of prognostic factors in the treatment of choanal atresia was conducted in 144 children (median age at first procedure at 8 months) with CA (77 had unilateral choanal atresia and 37 with bilateral choanal atresia). They identified the following risk factors for surgical failure: <6 mos of age, weight of <5 kg, and presence of bilateral choanal atresia. The authors also reported that the use of mitomycin, stenting, surgical approach, and presence of associated anomalies were not significantly associated with improvement in the surgical results [66,67].

Endoscopic bilateral CA repair in neonates is a safe procedure with a high long-term success rate [68]. Restenosis has been reported to occur frequently. Long-term follow-up (minimum one year) using nasal or rigid endoscopy, without systematic imaging is recommended by IPOG [26].

#### 5.2.6. Tracheostomy

Tracheostomy placement is generally reserved for neonates with multiple sites of airway obstruction and those who have failed non-surgical and other surgical options. Smith and colleagues reported that the mean duration of tracheostomy placement is 17 months in nonsyndromic children with RS vs. 32 months in syndromic children with RS. The majority of infants with tracheostomy will have a resolution of their OSA after mandibular growth [69]. Significant morbidities have been reported with tracheostomy which include recurrent infections, bleeding, granulomas, fistulas, tracheal stenosis, tracheomalacia, and accidental decannulation, which can be life-threatening [16].

The placement of a tracheostomy in an infant requires a significant number of resources in order to support the infant and family in contrast to adults. The need for specialized training for caregivers makes it also cumbersome and expensive along with significant complications with comorbidities.

## 6. Conclusions

OSA in neonates is often multifactorial and may arise from various conditions including prematurity, craniofacial malformations, neuromuscular diseases, and airway abnormalities. Clinical presentation is highly variable and often depends on the degree of airway compromise and presence of underlying syndrome. Polysomnography and direct visualization of the airway are essential in making the diagnosis. Treatment of neonatal OSA includes various surgical and non-surgical approaches and is typically dictated by the severity of respiratory and feeding difficulties. Our knowledge about the pathophysiology, treatment, and morbidity of pediatrics OSA has greatly evolved over the last decade. Unfortunately, most of the information is obtained from retrospective chart review as it is likely unethical to do randomized control trials in neonates or infants with OSA and the individual differences in pathophysiology, etc., make it difficult to compare case outcomes. Experience across different studies does suggest that a multidisciplinary approach for management may be best. The data looking at the long-term outcome of neonates with OSA also remain scarce. A prospective worldwide registry that enables long-term follow-up would be ideal to gather information about the trajectory of disease in neonates and infants and to fill gaps in the existing literature about various aspects related to neonatal OSA.

## Figures and Tables

**Table 1 children-09-00419-t001:** Difference in anatomy in Neonates vs. Older children.

Atomy	Neonates	Older Children	Implication of Neonatal Anatomy
Tongue	Relatively large in proportion to oral cavity	Normal	More prone to obstruction
Epiglottis	Long, floppy and omega shaped. Level C3–4	Firm, flatter Level C4–5	Prone to laryngomalacia and difficult intubation
Trachea	Smaller, shorter	Wider and longer	More prone to tracheomalacia and affects the critical closing pressure
Larynx Shape	Funnel shaped	Column	More prone to collapsibility and increased airway resistance
Narrowest point	Below glottis at level of cricoid cartilage	At level of vocal cords	Prone to tracheomalacia and consideration of the size of the required endotracheal tube
Airway Caliber	Smaller and shorter	Relatively wider and longer	Increased airway resistance and prone to obstruction
Occiput	Large	Normal	Prone to obstruction with neck flexion, reduced ability to clear secretion and positioning is important for intubation

**Table 2 children-09-00419-t002:** Conditions in Neonates commonly associated with OSA.

A. Prematurity B. Craniofacial abnormalities Maxillary hypoplasia Crouzon syndrome Apert syndrome Pfeiffer syndrome Goldenhar syndrome Trisomy 21 Achondroplasia Micrognathia and or retrognathia Treacher–Collins syndrome Pierre Robin sequence Stickler syndrome Macroglossia Trisomy 21 Beckwith–Wideman syndrome Achondroplasia C. Laryngeal abnormalities Laryngomalacia Airway hemangioma Subglottic stenosis Vocal cord paralysis D. Neurological/Neuromuscular Cerebral Palsy Congenital Myopathies (Nemaline Rod Myopathy, SMA) Myotonic Dystrophy Arnold–Chiari Malformation Central Hypotonia (Trisomy 21, Prader–Willi syndrome)

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
