# Peer review of "Obstructive Sleep Apnea in Neonates"

_children, 2022, doi:10.3390/children9030419_

Round 1

Reviewer 1 Report

Congratulations on your good review on OSA in neonates.

The suggestion:

a.

HFNC shoul be High flow Nasal Canula (HFNC)          (291)

b. References are requied 

  1. Pediatric OSA has ...1%-5% [ ],              (39)   
  2. Table 1                                                   (54)
  3.  RS                                                   (102)
  4. Table 2                                                   (65)
  5. Apert syndrome                              (112)
  6. Laryngomalacia                                     (119)
  7. Gastroesophageal reflux                        (120)
  8. NMD                                                       (124)
  9. Subglottic stenosis                                 (153)
  10. The severity scale of OSA                       (253)

Author Response

-

Reviewer 2 Report

Dear Authors,

You have compiled this subject, which requires a difficult and multidisciplinary approach, in an explanatory and sufficient way.  I suggest that, authors may cite more recent articles (mostly within the last 5-10 years).

Sincerly regards

Author Response

-